# Relationship between Apolipoprotein E Genotype and Lipoprotein Profile in Patients with Coronary Heart Disease

**DOI:** 10.3390/molecules27041377

**Published:** 2022-02-18

**Authors:** Yahui Lin, Qiong Yang, Zhaohui Liu, Baoman Su, Fen Xu, Yang Li, Jinsuo Kang, Zhou Zhou

**Affiliations:** Center of Laboratory Medicine, Beijing Key Laboratory for Molecular Diagnostics of Cardiovascular Disease, National Center for Cardiovascular Diseases & Fuwai Hospital, Peking Union Medical College & Chinese Academy of Medical Sciences, No.167, Beilishi Rd, Xicheng District, Beijing 100037, China; linyahui@fuwaihospital.org (Y.L.); yangqiongmed@163.com (Q.Y.); 15810084617@163.com (Z.L.); subm_fw@163.com (B.S.); xufen1224@163.com (F.X.); dyyang890701@163.com (Y.L.); kangjinsuo@fuwaihospital.org (J.K.)

**Keywords:** *APOE* genotype, lipoprotein profile, coronary heart disease, nuclear magnetic resonance, residual risk

## Abstract

(1) Background: Apolipoprotein E(ApoE) plays a critical role in lipid transport. The specific allele of *APOE* being expressed is associated with the development of coronary heart disease (CHD), however the specific mechanisms by which ApoE drives disease are unclear. In this study, we investigated the relationship between *APOE* allele, lipoprotein metabolome, and CHD severity to provide evidence for the efficacy of clinical cholesterol-lowering therapy; (2) Methods: Blood samples were collected from 360 patients with CHD that were actively being treated with statins. The lipoprotein profile, including particle numbers, particle size, and lipoprotein composition concentrates, was measured by nuclear magnetic resonance (NMR) spectroscopy. The severity of CHD was determined by quantifying coronary angiography results using the Gensini scoring system; (3) Results: We found there was no significant difference in low-density lipoprotein cholesterol (LDL-C) levels among ε2+ (ε2 allele carriers, consisting of ε2/ε2 and ε2/ε3 genotypes), ε3 (consisting of ε3/ε3 and ε2/ε4 genotypes), and ε4+ (ε4 allele carriers, consisting of ε3/ε4 and ε4/ε4 genotypes) participants receiving statin treatment. Compared with the ε3 group, patients with the ε2+ genotype showed lower concentrations of total low-density lipoprotein (LDL), small-LDL, and middle-LDL particles, as well as a larger LDL size, higher very low-density lipoprotein (VLDL) composition concentrates, and higher intermediate density lipoprotein (IDL) composition concentrates. The ε4+ group showed higher concentrations of total LDL, small LDL particles, and LDL compositions with smaller LDL size. The higher level of small LDL concentration was associated with a high Gensini score (B = 0.058, *p* = 0.024). Compared with the ε3 group, the risk of increased branch lesions in the ε2+ group was lower (OR = 0.416, *p* = 0.027); (4) Conclusions: The specific allele of *APOE* being expressed can affect the severity of CHD by altering components of the lipoprotein profile, such as the concentration of small LDL and LDL size.

## 1. Introduction

Recent studies have indicated that regulating levels of low density lipoprotein cholesterol (LDL-C), the main therapeutic target for treating coronary heart disease (CHD), cannot fully control disease, suggesting other factors are contributing to the risk of cardiovascular disease. Furthermore, patients labeled as low cardiovascular risk by conventional lipid parameters can still experience cardiovascular events [1,2]. When the LDL-C levels of 136,905 hospitalized coronary heart disease (CHD) patients were assessed, it was found that only half of the patients have elevated LDL-C levels (>100 mg/dL) [3]. Moreover, intervention studies on cholesterol-lowering therapeutics have shown that reaching optimal cholesterol levels was not enough to eliminate residual risk [4]. These findings have led to further researches on other potential contributors to CHD, such as lipoproteins, and scientists have confirmed that there are atherogenic effects associated with lipoprotein particles [5,6].

Nuclear magnetic resonance (NMR) spectroscopy and polyacrylamide gradient gel electrophoresis (GGE) are most commonly used to measure lipoprotein particles and their subclasses. The results of a prospective study showed that, although the concentration of lipoprotein particles can be detected by both GGE and NMR, NMR spectroscopy can more accurately predict the risk of CHD [7]. NMR spectroscopy has a variety of advantages, including being high throughput, having high accuracy and being highly repeatable [8].

It is widely acknowledged that *APOE* genotype can affect LDL-C levels and the risk of CHD. There are three alleles for *APOE* that encode three kinds of apolipoprotein E (ApoE), which have variations in their spatial conformation that impact lipid metabolism. ApoE exists on lipoprotein particles and plays an important role in cholesterol transport. The anti-atherosclerotic function of ApoE is largely due to its role in reducing plasma cholesterol levels by promoting the removal of triglyceride (TG)-rich lipoproteins from circulation. With regard to the effect of *APOE* genotype on lipoprotein particles, a longitudinal study in a Finnish population showed that the concentrations of very low-density lipoprotein (VLDL), intermediate density lipoprotein (IDL), low-density lipoprotein (LDL), and all LDL subclasses in *APOE* ε4 carriers were higher than those in the *APOE* ε3/ε3 genotype. They also found that the average particle size of LDL and HDL were smaller than those of ε3/ε3 genotype, which was consistent with their atherogenic effect. Meanwhile, *APOE* ε2 carriers have a larger LDL and HDL particle size [9]. Several other studies have also analyzed the relationship between *APOE* genotypes and lipoprotein particles in patients with sleep apnea, systemic lupus erythematosus, or special diets [10,11]. In 2017, the American Association of Clinical Endocrinologists (AACE) recommended ε4 as a non-traditional risk factor for dyslipidemia and atherosclerosis [12]. The purpose of our study is to use lipoprotein particle parameters detected by NMR spectroscopy to cross-sectionally analyze the relationship between expression of *APOE* alleles, lipoprotein particles, and disease severity in CHD patients being treated with statins, so as to provide additional evidence for the efficacy of cholesterol-lowering therapy.

## 2. Results

### 2.1. Baseline Clinical Characteristics

Overall, our study included 360 patients with CHD. Of these patients, 47 were carriers of *APOE* ε2/ε2 and ε2/ε3 genotypes (defined as ε2+ group), 252 were carriers of *APOE* ε3/ε3 and ε2/ε4 genotypes (defined as ε3 group). and 61 were carriers of *APOE* ε3/ε4 and ε4/ε4 genotypes (defined as ε4+ group). All clinical characteristics, except the serum levels of apo A1 and apo B, were compared across ε2+, ε3, and ε4+ and no significant difference was detected in LDL-C levels and other variables (Table 1). Notably, the serum level of apo A1 was found to be significantly lower in the ε4+ group compared to patients in groups with a ε2 and ε3 genotype (*p* < 0.05). The serum level of apo B in the ε4+ group was the highest among three groups (*p* < 0.05).

### 2.2. APOE Genotypes Influence on Lipoprotein Particle Concentrations

The individuals with different *APOE* genotypes showed significantly different lipoprotein particle levels when assessed for VLDL, IDL, and LDL subclasses. Appendix A showed that VLDL particle concentration was lower in the ε3 group and higher in the ε2+ group (*p* = 0.008). Patients who were carriers of the ε2+ genotype have less LDL particles, especially small-LDL and middle-LDL particles. Conversely, the concentration of LDL particles was higher in patients with the ε4+ genotype. We found no significant difference in IDL concentration across the three groups. The size of LDL particles was shown to be smaller in the ε4+ group and bigger in the ε2+ group (Figure 1).

The ε2+ genotype was shown to be independently associated with an increase in the concentration of VLDL (B = 49.51, *p* < 0.001), large LDL (L-LDL) (B = 39.9, *p* = 0.031), and LDL particle size (B = 0.236, *p* < 0.001), and a decrease in the concentration of LDL (B = −125.9, *p* = 0.036), small LDL(S-LDL) (B = −112.3, *p* < 0.001), and middle LDL(M-LDL) (B = −50.4, *p* = 0.041), when adding gender, age, BMI, diabetes, hemoglobin A1c (HbA1c), high-sensitivity C-reactive protein (hs-CRP), alanine aminotransferase (ALT) and serum creatinine (Cr) as covariates in our multiple linear regression model between *APOE* genotype and lipoprotein particle concentration. The relationship between the ε4+ genotype and a higher concentration of LDL (B = 142.07, *p* = 0.008) and S-LDL (B = 69.3, *p* = 0.037) are shown in Figure 2a.

### 2.3. Association between APOE Genotype and LDL Density Patterns

Next, we assessed the LDL particle pattern in the 360 CHD patients in our study, and we found that 303 patients displayed pattern B and 57 patients displayed pattern A (Table 2) [13,14]. In the ε2+ genotype patients, LDL pattern B was dominant, accounting for 68% of patients. In patients with the ε4+ genotype, an even higher 89% of patients displayed LDL pattern B. Figure 2b shows that the odds ratio of the ε2+ group with pattern B LDL is 0.316 (95%CI: 0.145~0.686, *p* = 0.004) compared with the ε3 group, indicating a decreased risk of CHD events, even after adjusting for gender, age, BMI, diabetes, HbA1c, hs-CRP, ALT, and serum creatinine.

### 2.4. Lipoprotein Compositions

The metabolome of various lipoproteins, including cholesterol, triglycerides, free cholesterol, phospholipids, apoA1, apoA2, and apoB100 were analyzed by NMR spectroscopy (Appendix A). Multiple linear regression analyses, after adjusting for confounding factors, demonstrated that the ε2+ genotype have relative higher risks of increased concentration of total triglycerides, VLDL 1–5, LDL-1 compositions, and the triglyceride of all HDL particles, while decreased levels of apoB100, cholesterol, free cholesterol, and phospholipid compositions of smaller LDLs (LDL 4, 5, and 6) were lower risks for CHD (Figure 3). Although the association between ε4+ genotype and lipoprotein compositions were not statistically significant, there was a tendency for patients with the ε4 + allele to have an increase in the concentration of middle and small LDL particle compositions (LDL 3, 4, 5, and 6).

### 2.5. Severity of Coronary Heart Disease

In our study cohort, we found that single-branch, double-branch, and three-branch lesions accounted for 32.5%, 41.0%, and 26.5% CHD, respectively. Figure 2b shows that the OR value of the ε2+ genotype was 0.416 (*p* = 0.027), indicating that the ε2+ genotype is associated with decreased branch lesions. The high Gensini score of all cases, indicating multivessel and severe CHD, was significantly associated with increased S-LDL particle concentration and LDL particle size, as determined by multiple linear regression analysis after adjusting for confounding factors (Table 3). The relationship between lipoprotein composition concentration and Gensini score was illustrated in Figure 4. The concentrations of total apoB100, including LDL-apoB100 and apoB100 from LDL2, LDL3, and LDL4, were shown to increase the relative risk of having a high Gensini score. The total cholesterol, including VLDL-cholesterol, HDL-cholesterol, and cholesterol from VLDL2, VLDL3, VLDL4, LDL3 and LDL4, were also found to be independent risk factors for CHD severity. The free cholesterol of IDL, LDL, VLDL2, VLDL3, LDL1, LDL2, LDL3 and LDL4, the triglyceride of IDL, VLDL2, VLDL3, LDL2, LDL4 and LDL5, and the phospholipid of IDL, LDL, VLDL2, VLDL3, LDL2, LDL3 and LDL4 were all related to an increase in the Gensini score.

## 3. Discussion

The present study investigated the relationship between *APOE* genotypes, lipoprotein profiles, and disease severity in CHD patients treated with statins. Our results indicate that, although there was no significant difference in LDL-C level among three *APOE* genotype groups being treated with statins, the concentration of lipoprotein particles and their compositions illustrated the different CHD risk between ε2+ and ε4+ groups. These findings suggest that, in addition to statins, individualized and intensive lipid-lowering therapies should be designed for ε4 carriers to further reduce residual cardiovascular risk caused by their abnormal lipoprotein metabolome.

Dyslipidemia is known to be a major risk factor for cardiovascular disease and has led to approximately 9.2 million CVD events in China over the last 20 years. The 2012 International Study on Dyslipidemia-Chinese Research data showed that among the 25,317 patients aged ≥ 45 years who received cholesterol-lowering drug treatment in 122 hospitals across 27 provinces in China for at least 3 months, 98.0% of the patients used monotherapy, among whom the coverage of statins was 88.9%. From this study, it was determined that 38.5% of the population did not meet their target LDL-C levels [15]. Moreover, targeting LDL-C-alone cannot meet the requirements of cholesterol-lowering therapies in the current era of precision medicine.

In our study, by comparing clinical baseline measurements across the three *APOE* genotype groups, we found there was no significant difference in the LDL-C levels among ε2+, ε3, and ε4+ groups, while the concentration of apoB was shown to be the highest in the ε4+ group and the lowest in the ε2+ group. Further, when we compared differences in the lipoprotein particles across the three genotype groups, we found that S-LDL accounted for about 50% of the total LDL particle concentration. When compared to the ε3 group, the ε2+ group had lower levels of total LDL, S-LDL, and M-LDL particle concentrations, lower risk of pattern B LDL and larger LDL size, reflecting a lower cardiovascular risk of ε2+ genotype. The ε4+ group showed higher concentrations of total LDL, S-LDL particles, and smaller LDL size, which oppositely reflects increased residual cardiovascular risk.

Recent studies have shown that, compared to LDL-C, LDL particles are better indicators of cardiovascular risk. When the levels of LDL-P and LDL-C are inconsistent, LDL-P is more likely to predict the risk of atherosclerosis than LDL-C [16,17]. The correlation between LDL-P and cardiovascular disease incidence is about twice as high as that of LDL-C, and its density pattern is also an independent risk factor for myocardial infarction/death [17]. Small LDL particles enter the blood vessel wall more easily and are not conducive to clearance from plasma. Small LDL particles also have longer subendothelial retention time, lower affinity with LDL receptors, and are more likely to be oxidized to form oxidized LDL, thus leading to a stronger atherosclerotic effect [18]. In terms of guidelines, LDL-P is also proposed to be particularly suitable for patients who meet their LDL-C and non-HDL-C target [19]. The AACE also recommended LDL particle concentration as an additional risk factor for dyslipidemia and atherosclerosis [20]. There are also many biomarkers discovery studies in atherosclerotic diseases using NMR spectroscopy to perform lipidomic analyses. Biomarkers such as LDL-TG, remnant-like particle cholesterol (RLP-C), VLDL-TG, HDL-apoA1, and others have been reported to add residual risk to CHD [21]. These studies reflect the important role of lipoprotein profiles in the treatment of dyslipidemia. Our study provided an overview of the lipoprotein profiles of a CHD population already treated by statins. The ε2+ and ε4+ groups displayed different profiles, with the ε2+ group tending to increase the TG composition while decreasing the TC composition, especially in small LDL, and the ε4+ group reversely tending to increase the TC composition, especially in small LDL. All lipid composition, when in specific lipoprotein subclasses, showed an association with increased severity of CHD. As mentioned in the introduction, ApoE is associated with removing TG-rich lipoproteins from circulation. Chylous granules from the intestine and VLDL from the liver are lipolysed by lipoprotein lipase (LPL) in circulation. The ApoE on the residual lipoprotein particles binds to the LDL receptor (LDLR), LDLR-related proteins, and heparan sulfate proteoglycans (HSPG) on the surface of hepatocytes, and the residual particles are swallowed and removed from circulation. Some VLDL residues are quickly removed, while others are further lipolysed and gradually converted into IDL and, finally, into LDL. Human ApoE exists in three common genotypes (*APOE* ε2, *APOE* ε3 and *APOE* ε4) and this polymorphism affects the disease risk of carriers. *APOE* ε3 is considered a parental form and is related to normal plasma cholesterol levels, while the function of *APOE* ε2 and *APOE* ε4 genotypes is altered and related to the occurrence of hyperlipidemia. The ability of *APOE* ε2 to bind to LDLR is impaired, resulting in poor removal of triglyceride-rich lipoprotein residues from plasma, thus leading to hypertriglyceridemia. The reason for the higher distribution of atherosclerotic lipoprotein cholesterol in the plasma of *APOE* ε4 carriers is due to the different distribution of apoε4 and *APOE* ε3 in the plasma, which means enhanced binding of *APOE* ε4 to VLDL. In summary, *APOE* gene polymorphisms changes the binding, uptake, and catabolism of apoB-containing lipoproteins in the liver, as well as intestinal cholesterol absorption, bile acid formation, and endogenous cholesterol synthesis [22].

The influence of *APOE* genotype on lipoprotein composition concentration is consistent with the mechanism of *APOE* gene polymorphisms associated with hyperlipidemia mentioned above, with ε2+ increasing the composition concentration of VLDL, IDL, and large LDL, and decreasing the composition concentration of small to medium LDL, whereas ε4+ increases the composition concentration of LDL. From the perspective of composition, the ε2+ group mainly affects triglyceride concentration whereas the ε4+ group mainly affects cholesterol and apoB100 levels.

Higher Gensini score are associated with an increase in S-LDL concentration and LDL size employing regression analysis, after adjusting for gender, age, diabetes, BMI, HbA1c, hs-CRP, ALT, and serum creatinine. Compared with the ε3 group, the risk of increasing branch lesions in the ε2+ group is much lower. Therefore, we speculate that *APOE* genotype might affect the severity of coronary artery disease by affecting the lipoprotein profile. Based on this, we suggest that, in addition to statins, clinicians should develop individualized and intensive lipid-lowering therapies for ε4 carriers to further reduce residual cardiovascular risk.

When comparing the difference in lipoprotein profiles among the three groups and the effect of genotype on branch lesions, except for S-LDL and LDL, other lipoprotein particle concentration and particle size of the ε4+ group showed an increased risk tendency compared with the ε3 group, however the difference was not statistically significant. It was considered as the result of cholesterol-lowering therapy with statins. Existing studies show that statins can reduce the particle concentration of the dominant LDL subclass [23,24,25]. As mentioned above, S-LDL accounted for about 50% of the total LDL particles, which meant the S-LDL was the dominant part of total LDL particles. Reducing the dominant small LDL subclass by statins may relatively increase the percentage of larger LDL particles and consequently increase LDL size. As the S-LDL concentration in the ε4+ group was the highest, the ε4+ group was more likely to be affected by statins, and as a result, the LDL density distribution will become more similar to that of the ε3 group.

There are still some limitations in this study. First, our study is a cross-sectional analysis, the type, dosage, and duration of statins used in our subjects is unclear, which may affect the analysis of lipoprotein particles. Second, researchers, such as Mackey and colleagues, have shown that after adjusting for LDL-P and HDL-P, HDL-C shows no negative correlation with carotid intima-media thickness (cIMT) or CHD, however HDL-P is still independently correlated with cIMT and CHD [26]. However, due to methodological limitations, it was not possible to obtain particle concentration data on HDL and its subclasses, thus failing to investigate additional relationships that may be occurring between *APOE* genotype, HDL particle profile, and disease severity. Finally, a more accurate scoring system or evaluation parameter could have been used to comprehensively describe the severity of CHD.

## 4. Materials and Methods

### 4.1. Study Population

Between November 2018 and March 2019, 360 patients were admitted to Fuwai Hospital for CHD (who were treated with statins). Patients with available *APOE* genotype information were selected after the following exclusion criteria: (1) diseases that affect lipoprotein metabolism: people with abnormal thyroid, liver, and kidney function; (2) using drugs that affect lipid levels, except for statins, and have used hormones or chemotherapy in the past 3 months; (3) patients with acute cardio-cerebrovascular disease, cardiac function grade II, III, or IV (NYHA grade), valvular heart disease, myocarditis, recent surgery or severe trauma, previous percutaneous coronary intervention, or coronary artery bypass grafting. A total of 299 patients underwent percutaneous coronary intervention for the first time after admission, whose coronary artery stenosis information was used to analyze the relationship between *APOE* and disease severity. Our study was approved by the Ethics Committee of Fuwai Hospital and all patients enrolled in the study provided signed informed consent.

### 4.2. Clinical Data Collection and Laboratory Measurements

Basic clinical information, including age, sex, body mass index (BMI), diabetes history, and other baseline data were collected. Fasting blood samples were collected in coagulation promoting tubes and EDTA-K2 anticoagulant tubes within 24 h after admission. After centrifuging blood samples immediately for 15 min at 1500 g, serum and plasma samples were divided into aliquots and stored at −80 °C until analysis within 3 days. Peripheral blood leukocytes were separated for genotyping. Clinical laboratory measurements, including total cholesterol (TC), triglyceride (TG), high density lipoprotein cholesterol (HDL-C), low density lipoprotein cholesterol (LDL-C), lipoprotein (a), serum creatinine, alanine transaminase (ALT), and hypersensitivity C reactive protein (hs-CRP) were detected in blood serum separated from the coagulation promoting tubes by a Beckman AU5811 automatic biochemical analyzer.

### 4.3. APOE Genotyping

*APOE* alleles were determined based on two single-nucleotide polymorphisms (SNPs, rs429358 and rs7412) at locus 19q13.31. There are three alleles of the *APOE* gene that are determined by a base pair at the two SNPs: ε2, ε3, and ε4, forming six *APOE* genotypes. The homozygote ε3/ε3 is the most prevalent and is associated with normal lipid concentrations. Thus, in our study, the ε3+ group was treated as a reference group and was set at the origin in our figures. The base pairs of ε3/ε3 at SNP rs429358 and rs7412 are TT and CC, respectively, and other genotypes include ε2/ε2 (TT/TT), ε2/ε3 (TT/TC), ε2/ε4 (TC/TC), ε3/ε4 (TC/CC), and ε4/ε4 (CC/CC). *APOE* genotypes were determined using TaqMan SNP genotyping assays with genomic DNA extracted from peripheral blood leukocytes.

### 4.4. Lipoprotein Profile Measured by NMR Spectroscopy

EDTA-K2 anticoagulant plasma was prepared for lipoprotein subclasses detection by Bruker 600MHz Avance III NMR spectrometer [8]. We adopted the commercial Bruker IVDr Lipoprotein Subclass Analysis (B.I.-LISA) method. Absolute concentrations in units of millimoles per liter were determined from NMR spectra using reference signal provided by ERETIC method (Electronic Reference To access In vivo Concentrations, which was built in Bruker IVDr platform) [27]. The chemical shift was initially calibrated to the methyl signal of trimethylsilyl propanoic acid (TSP) using Topspin 3.6.0 and calibrated the alanine doublet at 1.48 ppm subsequently. This method requires integration of the signals corresponding to the -CH3 and -CH2- groups from lipoproteins appearing in the 1D 1H general NMR profile spectrum at chemical shifts of 0.8 and 1.25 ppm. Finally, VLDL, LDL, and HDL particles were divided into VLDL 1–5, LDL 1–6, and HDL 1–4, respectively, from largest to smallest diameter. The particle concentrations of VLDL, IDL, LDL, LDL subclasses, and the average particle size of LDL (LDL size) were obtained for data analysis in this study. The lipid and apolipoprotein components of the lipoproteins were also available. The range of LDL particle size obtained by NMR spectrometry was 18.3~23.0 nm. Referring to previous studies, we divided the average LDL particle size into three groups [9]. Average LDL particle size (called LDL size) was between 18.3~19.8 nm and were called small LDL (S-LDL; including LDL-5 and LDL-6), 19.8~21.3 nm were considered medium LDL (M-LDL; including LDL-3 and LDL-4), and 21.3~23.0 nm were L-LDL (including LDL-1 and LDL-2).

### 4.5. Severity of Coronary Heart Disease

According to the results of coronary angiography, at least one epicardial coronary artery or its main branch diameter stenosis ≥50% was diagnosed as CHD. The severity of coronary artery lesions in 299 patients who received percutaneous coronary intervention for the first time were evaluated by Gensini score and the number of diseased branches. The results of coronary angiography were read by two interventional cardiologists and the degree of coronary artery stenosis was evaluated. The Gensini score was calculated by the product of the score corresponding to the degree of stenosis and the score corresponding to the location [28]. Patients were divided into three groups by single-branch lesion, two-branch lesions, and three-branch lesions according to the involvement of the left anterior descending artery, left circumflex artery, and right coronary artery. When the lesions involved the left trunk, the number of branches were counted as involving both the left anterior descending branch and the left circumflex branch.

### 4.6. Statistical Analysis

All analyses were conducted with R program. The continuous variables of normal distribution were expressed by mean ± standard deviation, and the differences between groups were analyzed by one-way analysis of variance, while the continuous variables of non-normal distribution were expressed by median (25th–75th percentile), and the differences between groups were analyzed by the Kruskal–Wallis test. The classified variables were expressed as percentage and χ^2^ test was used. Multiple linear regression was used to analyze the independent risk factors of lipoprotein particle concentration, LDL particle size, and Gensini score. Logistic regression was used to analyze the independent risk factors of different LDL patterns and diseased branches in patients with CHD. False discovery rate (FDR) was corrected. *p* < 0.05 was considered statistically significant.

## 5. Conclusions

Although there is no statistically significant difference in LDL-C amongst the ε2+, ε3, and ε4+ genotype groups in patients with CHD, lipoprotein profile, as measured by NMR spectroscopy, provided more information about the cardiovascular risks associated with different genotypes. The ε4+ group was shown to have higher concentrations of atherogenic LDL subclasses and compositions and the ε2+ group displayed the opposite. Our results indicate that ε4 allele carriers should be treated with individualized cholesterol-lowering therapies to further reduce cardiovascular risks.

## Figures and Tables

**Figure 1 molecules-27-01377-f001:**
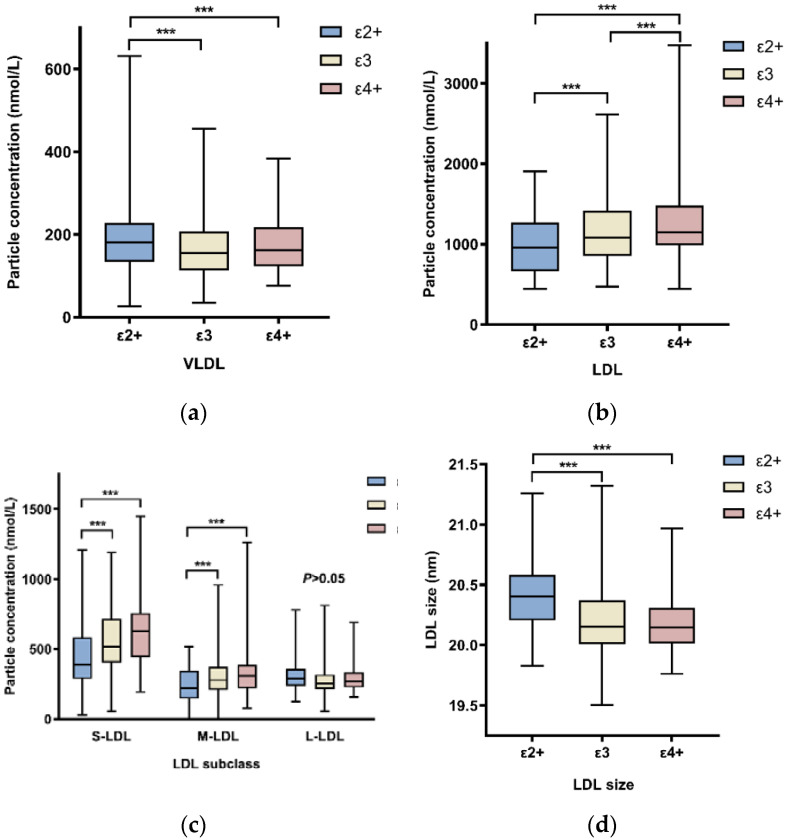
Comparison of lipoprotein particle concentrations and LDL size among three groups of differing *APOE* genotypes. One-way analysis of variance test was used to compare the means of continuous variables. (**a**) Comparison of VLDL particle concentrations. (**b**) Comparison of LDL particle concentrations. (**c**) Comparison of LDL subclass particle concentrations. (**d**) Comparison of LDL size. ***: *p*-value < 0.05. For clarity of illustration, only analysis results with *p* < 0.05 are shown here. Abbreviations: VLDL, very low-density lipoprotein; LDL, low-density lipoprotein; S-LDL, small low-density lipoprotein; M-LDL, median low-density lipoprotein; L-LDL, large low-density lipoprotein.

**Figure 2 molecules-27-01377-f002:**
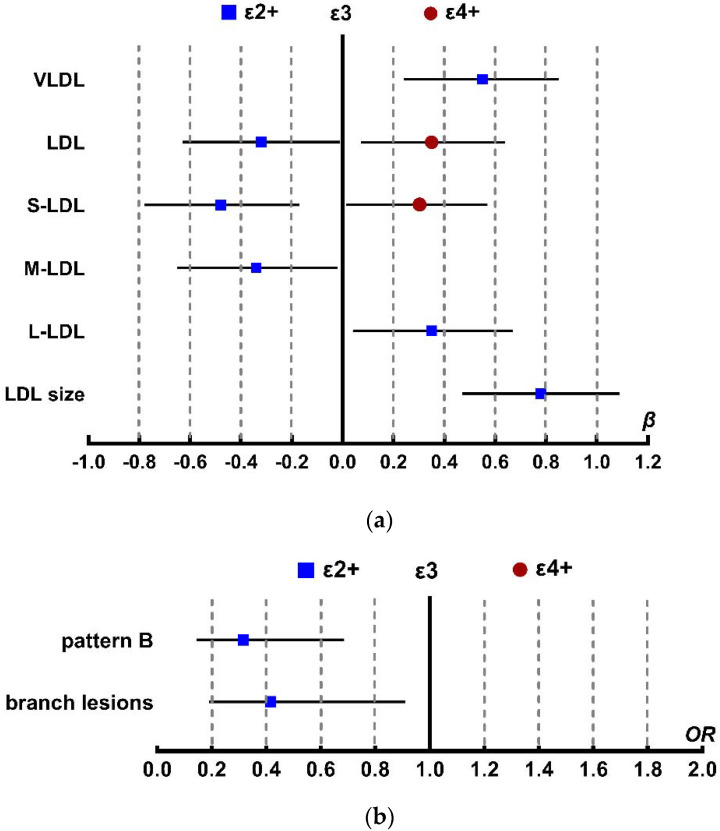
*APOE* affects the lipoprotein particle profile and the severity of CHD. Multiple linear regression and logistic regression models were adjusted for gender, age, diabetes, BMI, HbA1c, hs-CRP, ALT, and Cr. For clarity of illustration, only results of the analyses with *p* < 0.05 are shown here. (**a**) Multiple linear regression β-coefficients (x-axis)expressed by standard deviation (SD) units indicate the change in lipoprotein particle concentration over *APOE* genotype groups (ε2+, ε3, and ε4+). The most common ε3 group is set at the origin as a reference group and compared with ε2+ (squares) and ε4+ (circles) groups. Abbreviations: VLDL, very low-density lipoprotein; LDL, low-density lipoprotein; S-LDL, small low-density lipoprotein; M-LDL, median low-density lipoprotein; L-LDL, large low-density lipoprotein; (**b**) Logistic regression models are used to measure *APOE* effects on LDL density pattern and the severity of CHD.

**Figure 3 molecules-27-01377-f003:**
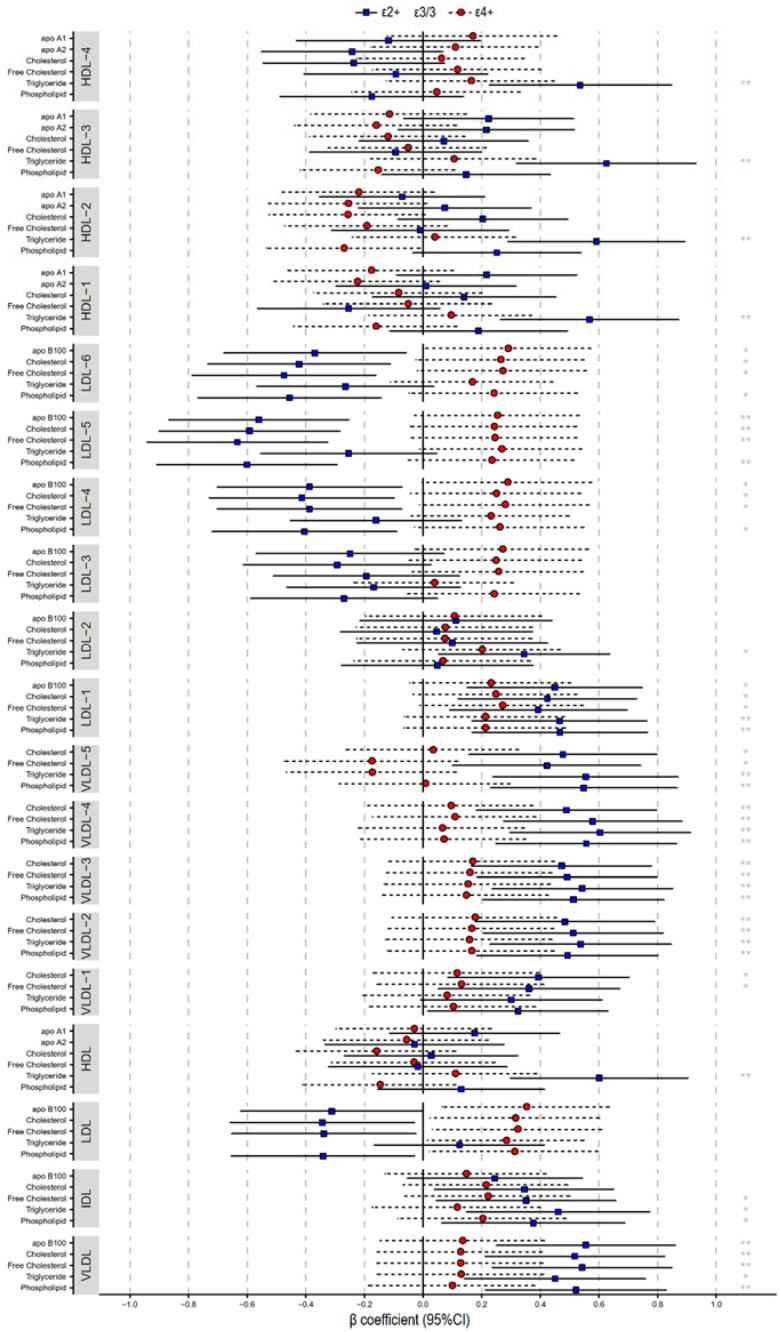
*APOE* affects the concentration of lipoprotein compositions. Multiple linear regression is adjusted for gender, age, diabetes, BMI, HbA1c, hs-CRP, ALT, and Cr. Multiple linear regression β-coefficients (x-axis) expressed by standard deviation (SD) units indicate the change in lipoprotein composition concentration over *APOE* genotype groups (ε2+, ε3, and ε4+). The most common ε3 group is set at the origin and compared with ε2+ (blue) and ε4+ (red) groups. *: *p*-value < 0.05; **: *p*-value < 0.01.

**Figure 4 molecules-27-01377-f004:**
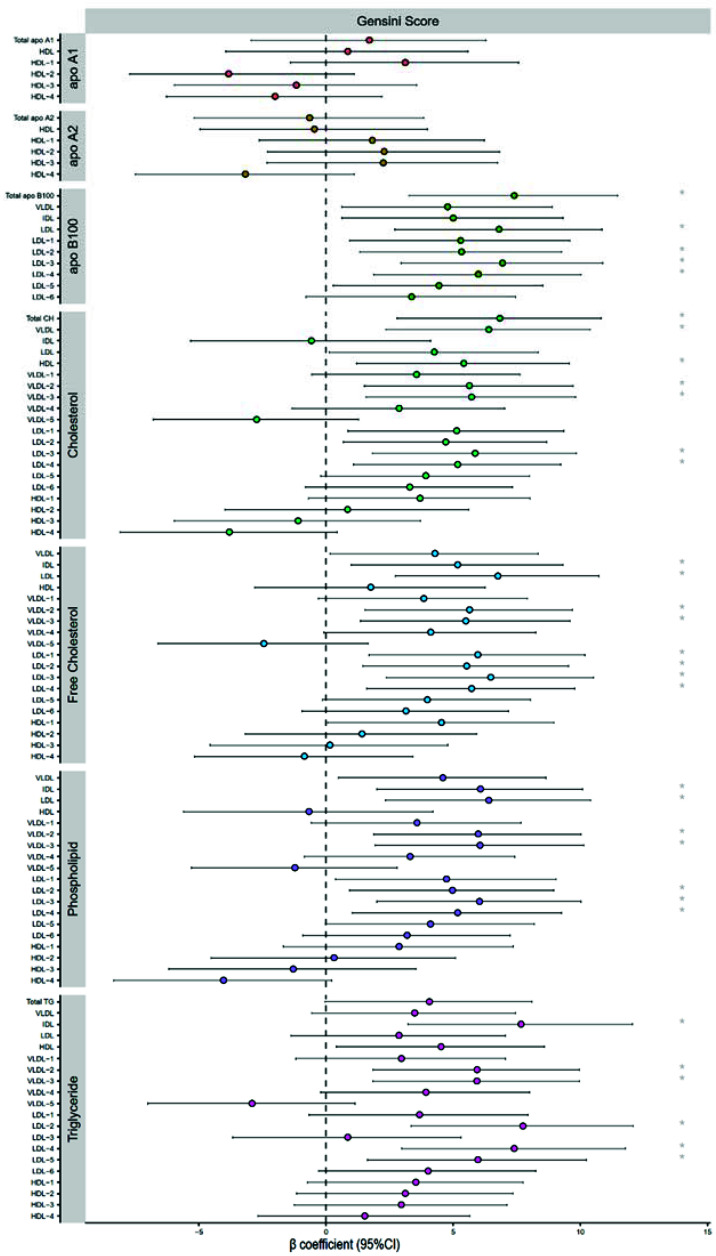
Multiple linear regression between lipoprotein composition concentration and Gensini score. Multiple linear regression β-coefficients (x-axis) expressed by standard deviation (SD) units indicate the change in Gensini score over lipoprotein composition concentrates. *: *p*-value < 0.05.

**Table 1 molecules-27-01377-t001:** The baseline characteristics of 360 patients grouped by *APOE* genotype.

	ε2+ (*n* = 47)	ε3 (*n* = 252)	ε4+ (*n* = 61)	*p* Value
Male (%)	25.00 (53.19)	138.00 (54.76)	38.00 (62.30)	0.446
DM (%)	12.00 (25.53)	86.00 (34.13)	23.00 (37.70)	0.394
Age (years)	59.17 ± 8.69	60.88 ± 9.34	61.57 ± 8.60	0.381
BMI (kg/m²)	26.10 (23.10, 28.35)	25.70 (23.70, 27.70)	26.00 (23.80, 27.70)	0.977
HbA1c (%)	6.00 (5.60, 6.90)	6.10 (5.70, 6.80)	6.00 (5.70, 6.80)	0.963
hs-CRP (mg/L)	1.34 (0.48, 3.50)	1.56 (0.51, 3.28)	0.99 (0.39, 2.63)	0.276
ALT (IU/L)	20.00 (15.00, 29.00)	21.00 (15.00, 31.00)	18.00 (12.50, 27.50)	0.206
Cr (μmol/L)	79.13 ± 19.41	77.71 ± 17.15	81.19 ± 16.44	0.360
TC (mmol/L)	4.25 ± 1.23	4.21 ± 1.05	4.34 ± 1.20	0.704
TG (mmol/L)	1.62 (1.18, 2.24)	1.45 (1.03, 1.90)	1.47 (1.18, 2.17)	0.074
HDL-C (mmol/L)	1.23 (1.01, 1.35)	1.18 (0.98, 1.36)	1.11 (0.96, 1.25)	0.064
LDL-C (mmol/L)	2.29 (1.46, 2.98)	2.27 (1.80, 3.00)	2.48 (2.04, 3.07)	0.254
apoA1 (g/L) *	1.49 ± 0.24	1.39 ± 0.33	1.31 ± 0.26	0.016
apoB (g/L) *	0.74 ± 0.23	0.78 ± 0.23	0.86 ± 0.28	0.034
Lp(a) (mg/L)	149.00(46.00,345.24)	212.00 (82.00, 443.07)	204.00 (100.64, 741.00)	0.055

* *p*-value < 0.05. Parameters are described by mean ± SD, *n*%, or median (IQR). Abbreviations: DM, diabetes mellitus; BMI, body mass index; HbA1c, glycated hemoglobin A1c; hs-CRP, hypersensitive C-reactive protein; ALT, alanine aminotransferase; Cr, creatinine; TC, total cholesterol; TG, triglycerides; HDL-C, high density lipoprotein cholesterol; LDL-C, low density lipoprotein cholesterol; apoA1, apolipoprotein A1; apoB, apolipoprotein B; Lp(a), lipoprotein(a).

**Table 2 molecules-27-01377-t002:** *APOE* genotypes and LDL density patterns in 360 patients.

	Pattern A	Pattern B	Total
ε2+	15	32	47
ε3	35	217	252
ε4+	7	54	61
Total	57	303	360

Pattern A, LDL size > 20.5 nm; Pattern B, LDL size ≤ 20.5 nm.

**Table 3 molecules-27-01377-t003:** Relationship between blood lipid concentrations and Gensini score.

	B	95%CI	*p*
cholesterol	−6.589	−30.084~16.907	0.581
triglyceride	5.517	−3.029~14.063	0.205
HDL-C	1.117	−26.503~28.737	0.937
LDL-C	2.099	−23.4~27.598	0.871
Lp(a) *	0.012	0~0.023	0.041
Small LDL *	0.058	0.008~0.108	0.024
Medium LDL	0.03	−0.011~0.071	0.154
Large LDL	−0.031	−0.109~0.047	0.438
LDL size *	42.254	2.193~82.315	0.039

* *p* < 0.05.

## Data Availability

The datasets generated and analyzed during the study are not publicly available due to legal reasons as the contained information could compromise research participant privacy and consent.

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
