# Peer review of "Relationship between Apolipoprotein E Genotype and Lipoprotein Profile in Patients with Coronary Heart Disease"

_molecules, 2022, doi:10.3390/molecules27041377_

Round 1

Reviewer 1 Report

The paper from Lin et al. shows interesting results correlating NMR-based lipoprotein profiles and clinical and genotypical parameters for CHD patients.
In spite of this ,the manuscript displays poor English language and lacks some specifications in the Materials and Methods sections.
Moreover, the discussion part can be improved, since for now it mostly resembles an introduction to CVD.
Improving these sections could then result in a good and interesting paper for the readers.

Minor comments
Lack of spaces between text and citations needs to be corrected.

ABSTRACT
APOE should be stated out at first use of the term.

INTRODUCTION
Lines 38-39: the analysis...showed.
Line 40: Moreover, intervention studies...showed.
Line 46: showed
Line 52: of being high throughput, having high accuracy and high repeatibility.
Line 61: have also analyzed

RESULTS
Lines 90-94 are best for the Materials and Methods section. 
Line 95: best to use passive voice and keep it consistent throughout the paper, so "The differences in total particle concentrations... were analyzed"
Line 133: ASCVD is never defined before, it needs to be stated out.
Figure 4: Gensini with capital G in x-axis. Also put a space between coefficient and the brackets.

DISCUSSION
Line 190: rephrase.
Line 200: Recent studies showed that, compared to LDL-C, LDL...
Line 205: no spaces between text and /
Line 208: space after .
Line 210: AACE is never defined
Line 236: Further, as differences of lipoprotein particles among the three groups aere compared, it is found...
Line 245-246: increasing...decreasing; wherease...increases.
Line 248-249: affects.
Line 262: the dominant LDL subclass
Line 264:", and reducing" or "thus reducing"
Line 270: researchers such as Mackey and colleagues
Line 273: it was not possible to obtain

MATERIALS AND METHODS
There is a lack of detail for NMR sample preparation and analysis. This needs to be included.
Line 279: Between November 2018 and March 2019
Line 280: who were treated with statins
Lines 284-285: roman numerals are in a different font, they need to be changed. 
Line 301: were divided
Line 305: were also available
Line 328: remove space before the square (2)

Author Response

Dear Reviewer,

Thank you for your kind and detailed suggestions. Please check the point-by-point response in the attachment.

Reviewer 2 Report

The manuscript “Relationship Between Apolipoprotein E Genotype and Lipo-2 protein Profile in Patients with Coronary Heart Disease” describes an NMR study of blood to analyse the relationship among APOE genotype, lipoprotein profile and disease severity in patients with coronary heart disease, aiming to provide more accurate evidence for clinical cholesterol-lowering therapy. The topic is interesting and important, but the results are difficult to understand and should be improved, as well as the materials & methods and the conclusions.

MAJOR CONCERNS:

Page 2, lines 71-74: How were the genotypes determined in the 360 patients?

Page 4, line 112: How the adjustment for male, age, diabetes, BMI, etc, was performed?

Figures: How should the reader interpret Figures 2, 3 and 4? What is the threshold of multiple linear regression β-coefficients to consider a significant effect?

Table 3: What is the meaning of B and CI?

Page 11, line 299: Details on sample collection (volume, centrifugation conditions, etc) and storage (temperature and how long) until analysis should be provided

Page 11, lines 300-304: Sample preparation protocol, NMR pulse sequence, and NMR acquisition parameters must be provided. Detailing information regarding the methods used for lipoprotein distribution and quantification are also missing

Page 11, line 322: In statistical analysis, p-values should be corrected for multiple comparisons using Bonferroni or FDR correction

Page 11, line 333: Conclusions must be improved to elucidate the main findings of this study

Supplementary information: the meaning of asterisks in Tables is missing. When it reads 0.000 in P correct to <0.001

MINOR CONCERNS:

Abstract: define the meaning of ε2+, ε3 and ε4+ groups

Page 1, Line 21; page 2, line 90: NMR spectrometry instead of NMR spectrometer

Figures 2 and 3: Please indicate the meaning of x-axis in figure

Author Response

Dear Reviewer,

Thank you for your constructive and detailed suggestions. Please check the point-by-point response in the attachment.

Round 2

Reviewer 2 Report

Authors addressed all reviewer concerns and improved significantly the manuscript.